# Olive Leaf Extract Supplementation to Old Wistar Rats Attenuates Aging-Induced Sarcopenia and Increases Insulin Sensitivity in Adipose Tissue and Skeletal Muscle

**DOI:** 10.3390/antiox10050737

**Published:** 2021-05-07

**Authors:** Daniel González-Hedström, Teresa Priego, Sara Amor, María de la Fuente-Fernández, Ana Isabel Martín, Asunción López-Calderón, Antonio Manuel Inarejos-García, Ángel Luís García-Villalón, Miriam Granado

**Affiliations:** 1Departamento de Fisiología, Facultad de Medicina, Universidad Autónoma de Madrid, 28029 Madrid, Spain; dgonzalez@pharmactive.eu (D.G.-H.); sara.amor@uam.es (S.A.); maria.delafuente@uam.es (M.d.l.F.-F.); angeluis.villalon@uam.es (Á.L.G.-V.); 2Pharmactive Biotech Products S.L. Parque Científico de Madrid, Avenida del Doctor Severo Ochoa, 28049 Madrid, Spain; aminarejos@hotmail.com; 3Departamento de Fisiología, Facultad de Medicina, Universidad Complutense de Madrid, 28029 Madrid, Spain; tpriegoc@med.ucm.es (T.P.); anabelmartin@med.ucm.es (A.I.M.); alc@med.ucm.es (A.L.-C.); 4CIBER Fisiopatología de la Obesidad y Nutrición, Instituto de Salud Carlos III, 28029 Madrid, Spain

**Keywords:** aging, adipose tissue, skeletal muscle, sarcopenia, insulin resistance, olive leaf extract, inflammation, oxidative stress

## Abstract

Aging is associated with increased visceral adiposity and a decrease in the amount of brown adipose tissue and muscle mass, known as sarcopenia, which results in the development of metabolic alterations such as insulin resistance. In this study, we aimed to analyze whether 3-week supplementation with a phenolic-rich olive leaf extract (OLE) to 24 months-old male Wistar rats orally (100 mg/kg) attenuated the aging-induced alterations in body composition and insulin resistance. OLE treatment increased brown adipose tissue and attenuated the aging-induced decrease in protein content and gastrocnemius weight. Treatment with OLE prevented the aging-induced increase in the expression of PPAR-γ in visceral and brown adipose tissues, while it significantly increased the expression of PPAR-α in the gastrocnemius of old rats and reduced various markers related to sarcopenia such as myostatin, HDAC-4, myogenin and MyoD. OLE supplementation increased insulin sensitivity in explants of gastrocnemius and epididymal visceral adipose tissue from aged rats through a greater activation of the PI3K/Akt pathway, probably through the attenuation of inflammation in both tissues. In conclusion, supplementation with OLE prevents the loss of muscle mass associated with aging and exerts anti-inflammatory and insulin-sensitizing effects on adipose tissue and skeletal muscle.

## 1. Introduction

Aging is associated with changes in body composition that are characterized by a significant decrease in muscle mass and an increase and redistribution of fat mass that is mainly concentrated in the central area of the body [1]. These changes are related to functional decline and may predict mortality due, at least in part, to the increase in the waist circumference and waist-to-hip ratio, which is associated with an increased risk of developing cardiovascular diseases [2]. 

The loss of muscle mass associated with aging, known as sarcopenia, implies fragility and a decrease in strength that, in the long term, can lead to disability and loss of independence [3]. The aging-induced physiological and morphological changes in skeletal muscle are characterized by marked infiltration of fibrous and adipose tissues and an overall decline in the number and size of muscle fibers, particularly the type 2 or fast-twitch muscle fibers [3]. The loss of muscle fibers is the result of both an increase in muscle proteolysis and a decrease in protein synthesis [4], as well as a significant decline in the number of neuromuscular junctions [5,6]. Different systemic factors are involved in these processes, such as alterations in the circulating levels of certain hormones such as growth hormone, IGF-I, glucocorticoids, sex hormones, as well as serum levels of pro-inflammatory cytokines such as tumor necrosis factor-alpha (TNF-α) or interleukin-6 (IL-6) [7]. There are also alterations in the expression of local factors involved in protein synthesis, such as IGF-I and its binding proteins [8], or markers involved in the proteolytic process, such as the atrogenes MURF-1 or atrogin-1 [9]. Likewise, sarcopenia associated with aging causes an increase in the expression of myostatin [10], a protein that inhibits the progression of the cell cycle and regulates the levels of factors that control muscle growth [11]. Myostatin negatively impacts muscle metabolism through inhibition of AMPK kinase and other effectors, which results in impaired mitochondrial biogenesis, insulin sensitivity and beta-oxidation, promoting the development of muscle lipotoxicity and apoptosis of muscle fibers [12].

Deterioration of muscle mass in aging is also influenced by alterations in muscle regeneration, a process that is mediated by factors produced by satellite cells such as MyoD or myogenin [13]. Another factor that seems to play a major role in sarcopenia is HDAC-4, a class IIa HDAC that deacetylates myosin heavy chains, PGC-1α, and Hsc7 to control muscle homeostasis [14]. This protein is highly expressed in sarcopenic muscles from both animals [15,16] and humans [17], and its upregulation is related with the underlying loss of neuromuscular function.

Aging not only affects skeletal muscle but also the structure and function of adipose tissue. Overall, aging is associated with an increase in visceral adiposity [18] due to a chronic positive energy balance, a decrease in catecholamine-induced lipolysis in visceral adipose tissue [19] and a shift in lipid storage from the subcutaneous to the visceral fat depot [18]. Furthermore, a reduction in the amount of brown adipose tissue is also present in aged individuals [20], with this fact being associated with an impaired thermoregulatory capacity [21]. 

The secretory profile of adipose tissue is also influenced by aging, with increased secretion of leptin and proinflammatory cytokines such as IL-1β, IL-6 and TNF-α and decreased secretion of anti-inflammatory adipokines such as adiponectin [22]. 

A common phenomenon that affects both adipose tissue and skeletal muscle in aging is insulin resistance, which is associated with both the oxidative status [23] and the chronic low-grade inflammatory state associated with this condition [24]. The greater production of reactive oxygen species (ROS) and the increase in the local production of pro-inflammatory cytokines impair the activation of the PI3K/Akt insulin signaling pathway, limiting the expression and translocation of GLUT-4 transporters to the cell membrane, and therefore reducing glucose uptake [25,26]. The lower glucose uptake from the extracellular medium contributes to increasing circulating glucose levels, producing hyperglycemia. To compensate for the impaired insulin sensitivity, β-cells secrete higher amounts of insulin, leading to hyperinsulinemia, a condition that has deleterious effects on cardiovascular function due to the proliferative effects of insulin on vascular smooth muscle cells, which produces thickening of wall arteries. Moreover, hyperinsulinemia has a strong impact on lipid metabolism producing hypertriglyceridemia and increasing endogenous lipid synthesis and cholesterol transport into arteriolar smooth-muscle cells, enhancing the atherogenic process [27].

In adipose tissue, impaired insulin signaling not only affects glucose uptake but also lipid storage, favoring the intracellular deposition of lipids in other tissues such as the liver or the skeletal muscle [28]. 

In skeletal muscle, insulin resistance is closely related to the phenomenon of sarcopenia. Indeed, an increase in muscle mass in aged rats rises skeletal muscle glucose uptake and improves insulin sensitivity [29]. On the other hand, myostatin blockade in the muscle of aged animals not only prevents sarcopenia but also increases insulin sensitivity [30]. 

In the last decades, the number of elderly people has increased exponentially in both developed and developing countries, and it is expected that this trend will continue to rise in the coming years [31]. This makes necessary the search for new preventive and therapeutic strategies that help to alleviate the alterations associated with aging, if possible, with greater accessibility and fewer side effects than conventional pharmacological treatments. Among them, nutraceutical products and functional foods arise as good candidates and constitute a great promise to improve health and prevent aging-related chronic diseases [32].

The Mediterranean diet is reported to be effective at extending the life span and reducing some of the comorbidities associated with aging [33]. This diet is especially rich in products derived from the olive tree, mostly olive oil [34], which attenuates inflammation and the underlying development of insulin resistance and sarcopenia [35]. In addition to olive oil, there are other products derived from olive trees, such as olive leaf extracts (OLE), that are also very interesting from a biological point of view due to their high content of phenolic compounds. OLE exerts several health benefits such as antioxidant, antihypertensive, hypoglycemic, hypocholesterolemic, anti-inflammatory and anti-obesity [36]. Despite its multiple beneficial effects, there are very few studies in which its usefulness has been assessed for the prevention and/or treatment of metabolic disorders associated with aging, mainly insulin resistance. In this sense, a very recent study of our group has shown how the administration of OLE to old rats for 3 weeks improves hepatic insulin sensitivity and attenuates endothelial dysfunction and insulin vascular resistance [37]. However, the effects of OLE on aging-induced sarcopenia and insulin resistance in adipose tissue and skeletal muscle have not been studied yet. Thus, the aim of this work is to analyze whether the supplementation of old Wistar rats with an extract of olive leaves for 21 days decreases sarcopenia and insulin resistance.

## 2. Materials and Methods

### 2.1. Materials

Pharmactive Biotech products S.L. (Madrid, Spain) provided water-soluble samples of olive leaf extract (OLE) from *Olea europaea* L. standardized in 30% of ortho-diphenols by UV/Vis and in 1 mg/g of luteolin-7-*O*-glucoside by HPLC. All samples were in powder and were stored in darkness until their addition into the feeding bottles. The extract composition has been previously described with secoiridoids, as elenolic acid derivatives of simple phenolic compounds (hydroxytyrosol, oleacein, oleuropein or ligustroside) and flavonols (quercetin 3-*O*-rutinoside and quercetin-3-*O*-glucoside, luteolin, diosmin or apigenin) being identified by RP-HPLC-PAD-MS [37].

### 2.2. In Vivo Study

#### 2.2.1. Animals

Three-month-old (Young; *n* = 11) and 24-month-old (Old; *n* = 14) male Wistar rats were housed with access to standard chow and water *ad libitum* under controlled conditions of humidity (50–60%) and temperature (22–24 °C). All the experiments and handling of animals were conducted according to the European Union Legislation and with the approval of the Animal Care and Use Committee of the Universidad Autónoma de Madrid and the regional authorization from the Government of Comunidad de Madrid (*PROEX 048*/*18*).

#### 2.2.2. Treatment

Half of the old rats received drinking water ad libitum (Old), and the other half were treated for 21 days with 100 mg/kg of OLE dissolved in the drinking water. The solution was renewed every 3 days, and water intake was assessed in order to ensure a correct dosage depending on the volume ingested by rats.

Their body weight and food intake were controlled daily over the treatment period. After the 21 days of treatment, glycemia was measured after overnight fasting by venous tail puncture using the glucometer Glucocard G^TM^ (Arkray Factory, Inc., Koji Konan-cho, Koka, Shiga, Japan) and killed by decapitation under an overdose of sodium pentobarbital (100 mg/Kg i.p.). To obtain the serum, trunk blood was collected and centrifuged at 3000 rpm for 20 min. Epididymal visceral, lumbar subcutaneous, interscapular brown and aortic perivascular adipose tissue depots, as well as the soleus and gastrocnemius muscles, were removed and weighed after the sacrifice and stored at –80 °C for further analysis.

#### 2.2.3. Incubation in Presence/Absence of Insulin (10^−7^ M) of Epididymal Adipose Tissue and Gastrocnemius Muscle Explants 

In an incubator with 95% O_2_ and 5% CO_2_ at 37 °C_,_ 100 mg explants of epididymal adipose tissue and gastrocnemius muscle were incubated in 1.5 mL of 1:1 mixture of Dulbecco’s Modified Eagle’s Medium and Ham’s F-12 medium (DMEM/F-12) with Gibco’s glutamine (Invitrogen, Carlsbad, CA, USA) in the presence/absence of insulin (10 ^−7^ M) (Sigma-Aldrich, St. Louis, MO, USA). The medium was supplemented with 100 U/mL penicillin and 100 μg/mL streptomycin (Invitrogen, Carlsbad, CA, USA). After incubation, explants were collected and stored at –80 °C for further analysis.

#### 2.2.4. Western Blot

100 mg of epididymal adipose tissue and gastrocnemius muscle of each animal were homogenized using RIPA buffer. After centrifugation for 20 min at 12,500 rpm and 4 °C, the total protein content in the supernatant was analyzed by the Bradford method (Sigma-Aldrich; St. Louis, MO, USA) [38]. For the electrophoresis, resolving gels with SDS acrylamide (10%) (Bio-Rad; Hercules, CA, USA) and 100 μg of protein in each well were used. Proteins were transferred to polyvinylidene difluoride (PVDF) membranes (Bio-Rad; Hercules, CA, USA), and Ponceau red dyeing (Sigma-Aldrich; St. Louis, MO, USA) was used to assess transfer efficiency. Tris-buffered saline (TBS) containing 5% (*w*/*v*) of non-fat dried milk for non-phosphorylated proteins or with 5% (*w*/*v*) bovine serum albumin (BSA) for phosphorylated proteins were used for membrane blockade. After the blockade, membranes were incubated overnight with the appropriate primary antibody (Akt 1:1000; Merck Millipore; Darmstadt, Germany and p-Akt (Ser473) 1:500; Cell Signaling Technology; Danvers, MA, USA). Then, the membranes were subsequently washed and incubated with the secondary antibody conjugated with peroxidase (1:2000; Pierce; Rockford, IL, USA) for 90 min to measure peroxidase activity by chemiluminescence, using the BioRad Molecular Imager ChemiDoc XRS System (Hercules, CA, USA). All data are referred to % values from young rats on each gel.

#### 2.2.5. RNA Extraction and Purification

Following the Tri-Reagent protocol [39], total RNA was extracted from gastrocnemius, epididymal white and interscapular brown adipose tissues and quantified with a Nanodrop 2000 (Thermo Fisher Scientific, Hampton, NH, USA). Of the total RNA, 1 µg was used to synthesize the cDNA by a high capacity cDNA reverse transcription kit (Applied Biosystems; Foster City, CA, USA).

#### 2.2.6. Quantitative Real-Time PCR

The mRNA levels of cyclooxygenase-2 (COX-2), tumoral necrosis factor-alpha (TNF-α), interleukin 1β (IL-1β), interleukin 6 (IL-6), inducible Nitric Oxide Synthase (iNOS), interleukin 10 (IL-10), glutathione peroxidase (GPx), glutathione reductase (GSR), superoxide dismutase-1 (SOD1), NADPH oxidases 1 and 4 (NOX1 and NOX4) and lipoxygenase (Alox5) were assessed in gastrocnemius and white adipose tissue by quantitative real-time polymerase chain reaction (qPCR). The insulin receptor (IR), glucose transporter 4 (GLUT-4), leptin receptor (Ob-R), carnitine palmitoyltransferase 1a (CPT-1a), β3 adrenergic receptor (Adrβ3), lipoprotein lipase (LPL), hormone-sensitive lipase (HSL), fatty acid synthase (FASn), peroxisome proliferator-activated receptor-gamma (PPARγ) and uncoupling protein 1 (UCP-1) gene expression were measured in the epididymal white and subscapular brown adipose tissues. In gastrocnemius, the mRNA levels of histone deacetylase 4 (HDAC-4), myogenin, myoblast determination protein (MyoD), atrogin-1, muscle RING-finger protein-1 (MuRF1), myostatin, insulin growth factor 1 (IGF-1), IGF-1 receptor, IGF-binding protein 5 (IGFBP-5), peroxisome proliferator-activated receptor α (PPARα), PPAR-γ coactivator 1α (PGC-1α), BCL2/adenovirus E1B 19 kDa protein-interacting protein 3 (Bnip-3) and microtubule-associated protein 1A/1B-light chain 3 (LC-3) were also measured. The amplification was performed in a Step One machine (Applied Biosystems, Foster City, CA, USA) using assay-on-demand kits (Applied Biosystems, Foster City, CA, USA) and TaqMan Universal PCR Master Mix (Applied Biosystems, Foster City, CA, USA). To determine relative expression levels, the ∆∆C_T_ method was used [40], and the values were normalized to the housekeeping gene 18S.

#### 2.2.7. Statistical Analysis

The values are expressed as means ± standard error of the mean (SEM) and analyzed by one-way ANOVA followed by Bonferroni post hoc test using GraphPad Prism 5.0. (San Diego, CA, USA). A *p*-value of < 0.05 was considered significant.

## 3. Results

### 3.1. Adipose Tissue Depots

The relative weights of epididymal visceral and lumbar subcutaneous adipose tissues as well as interscapular brown adipose tissue are shown in Table 1. Aging did not modify the amount of interscapular brown adipose tissue, but it induced a significant increase in the relative weights of both epididymal and lumbar adipose tissues (*p* < 0.001 for both). OLE treatment to old rats did not affect the amount of white adipose tissue, but it significantly increased the relative weight of brown adipose tissue (*p* < 0.05).

### 3.2. mRNA Levels of Lipid Metabolism Related Enzymes in White Adipose Tissue

The gene expression of leptin receptor (Ob-R), β3 adrenergic receptor (Adr β3), lipoprotein lipase (LPL), hormone-sensitive lipase (HSL), fatty acid synthase (FASn), peroxisome proliferator-activated receptor-γ (PPAR-γ) and un-coupling protein 1 (UCP-1) in epididymal white adipose tissue of young rats, old rats and old rats treated with OLE are shown in Figure 1. 

The mRNA levels of UCP-1 were unchanged between young and untreated old rats. However, the gene expression of LPL (*p* < 0.01), HSL (*p* < 0.05) and FASn (*p* < 0.05) were downregulated in response to aging, whereas the gene expression of PPAR-γ was significantly increased in aged rats compared to young ones (*p* < 0.05). 

OLE treatment to old rats downregulated the expression of Adrβ3 (*p* < 0.01) and attenuated the aging-induced changes in the mRNA levels of IR and PPAR-γ (*p* < 0.05 for both).

### 3.3. mRNA Levels of Lipid Metabolism and Thermogenesis-Related Enzymes in Brown Adipose Tissue

Figure 2 shows the mRNA levels of different proteins involved in lipid metabolism and thermogenesis in brown adipose tissue. 

Aged rats showed a significant increase in the gene expression of CPT-1a (*p* < 0.05), HSL (*p* < 0.001) and PPAR-γ (*p* < 0.05) compared to young animals. However, aging did not modify the mRNA levels of Ob-R, Adr β3, LPL and UCP-1 and significantly downregulated the gene expression of FASn (*p* < 0.01).

OLE supplementation to old rats downregulated the expression of Adr β3 (*p* < 0.05) and UCP-1 (*p* < 0.01) and attenuated the aging-induced increase in the mRNA levels of PPAR-γ (*p* < 0.05).

### 3.4. Body Weight and Protein Content in Soleus and Gastrocnemius Muscles

Aging was associated with a significant decrease in both gastrocnemius muscles (Figure 3A; *p* < 0.001) and soleus weights (Figure 3B; *p* < 0.001) and with a decreased protein content only in gastrocnemius muscle (Figure 3C; *p* < 0.001). 

OLE supplementation to old rats for three weeks did not induce any changes in soleus weight and total protein levels, but it attenuated the aging-induced decrease in protein content (*p* < 0.001) and muscle mass (*p* < 0.05) in the gastrocnemius.

### 3.5. Gene Expression of Sarcopenia-Related Markers in Gastrocnemius Muscle

The mRNA levels of several proteins involved in sarcopenia are shown in Figure 4.

Aging did not modify the gene expression of IGF-I and PPAR-α in gastrocnemius muscle but it significantly upregulated the mRNA levels of HDAC-4 (Figure 4A; *p* < 0.001), MyoD (Figure 4A; *p* < 0.01), myogenin (Figure 4A; *p* < 0.001), atrogin-1 (Figure 4A; *p* < 0.05), MURF-1 (Figure 4A; *p* < 0.01), Myostatin (Figure 4A; *p* < 0.05), IGF-1 receptor (Figure 4B; *p* < 0.01), IGFBP-5 (Figure 4B; *p* < 0.05) and LC3 (Figure 4B; *p* < 0.01). On the contrary, the gene expression of PGC-1α and Bnip-3 were downregulated in response to aging (Figure 4B; *p* < 0.05 for both). The treatment with OLE did not prevent the aging-induced alterations in the gene expression of atrogin-1, MURF-1, IGF-I receptor, IGFBP-5, Bnip-3 and LC3 but it significantly reduced the mRNA levels of IGF-I and attenuated the aging-induce alterations in the mRNA levels of HDAC-4, MyoD, myogenin, myostatin and PGC-1α (*p* < 0.05 for all).

### 3.6. Gene Expression of Oxidative Stress and Inflammatory Markers in Gastrocnemius Muscle

The mRNA levels of the proinflammatory markers COX-2 (Figure 5A; *p* < 0.05), IL-6 (Figure 5A; *p* < 0.01) and IL-1β (Figure 5A; *p* < 0.05) were increased in old untreated rats compared to young animals, whereas the gene expression of the anti-inflammatory cytokine IL-10 was significantly decreased (Figure 5A; *p* < 0.05). 

OLE administration prevented the aging-induced alterations in the gene expression of COX-2, IL-6, IL-1β and IL-10 (*p* < 0.05 for all). Neither aging nor OLE modify the gene expression of iNOS and TNF-α (Figure 5A).

The mRNA levels of SOD-1 and NOX-1 were decreased in response to aging (*p* < 0.05 for both), whereas the gene expression of GPx was significantly upregulated (*p* < 0.001). Treatment with OLE did not affect the gene expression of SOD-1 and NOX-1, but it significantly reduced the gene expression of Alox5 and attenuated the aging-induced upregulation of GPx (*p* < 0.05). The gene expression of GSR and NOX-4 was modified neither by aging nor by OLE administration.

### 3.7. Insulin-Induced Activation of PI3K/Akt Pathway and mRNA Levels of IR and GLUT-4 in Visceral Adipose Tissue and Gastrocnemius Muscle

OLE treatment attenuated the aging-induced decrease in the mRNA levels of GLUT-4 in the gastrocnemius (Figure 6B, *p* < 0.05) and of IR in the epididymal visceral adipose tissue (Figure 6D, *p* < 0.05).

No changes were found among experimental groups in the pAkt/Akt ratio in basal conditions neither in gastrocnemius (Figure 6A) nor in epididymal adipose tissue (Figure 6B) explants. In gastrocnemius explants, incubation with insulin for 15 min significantly upregulated the pAkt/Akt ratio in young and old rats treated with OLE (*p* < 0.05 for both) but not in untreated aged rats. In explants of epididymal adipose tissue, insulin stimulation also increased the pAkt/Akt ratio in young rats (*p* < 0.05) but failed to activate the PI3K/Akt pathway in untreated old rats. However, explants from old animals treated with OLE showed higher activation of this pathway in response to insulin (*p* < 0.05). 

### 3.8. Gene Expression of Oxidative Stress and Inflammatory Markers in Visceral Epididymal Adipose Tissue

No significant changes were found among experimental groups in the mRNA levels of iNOS, TNF-α and IL-6 (Figure 7A). However, the mRNA levels of the proinflammatory marker IL-1β were significantly increased in old untreated rats compared to young animals (Figure 7A; *p* < 0.001), whereas the gene expression of COX-2 (Figure 7A; *p* < 0.001) and IL-10 (Figure 7A; *p* < 0.05) were downregulated in response to aging. Treatment with OLE did not modify the aging-induced changes in the mRNA levels of COX-2, but it prevented the alterations in the mRNA levels of IL-1β and IL-10 (*p* < 0.05 for both).

The gene expression of GPx was not affected neither by aging nor by OLE treatment (Figure 7B). However, the gene expression of GSR (*p* < 0.01), SOD-1 (*p* < 0.01), NOX-1 (*p* < 0.01), NOX-4 (*p* < 0.05) and Alox-5 (*p* < 0.01) was significantly decreased in response to aging (*p* < 0.01 for all). Treatment with OLE did not change the mRNA levels of GSR and SOD-1 but it attenuated the aging-induced downregulation of NOX-1 (*p* < 0.05). and NOX-4 (*p* < 0.05).

## 4. Discussion

In this paper, we report for the first time the beneficial effects of an olive leaf extract rich in flavonoids and secoiridoids, administered at a dose of 100 mg/kg/day for three weeks, on sarcopenia and insulin sensitivity in skeletal muscle and adipose tissue of aged rats. This dose is considerably lower than the one used by other authors in an experimental model of aging in rats (100 mg/kg vs 500 and 1000 mg/kg) [41] and was selected based on its positive effects for the alleviation of several inflammatory conditions in rodents such as sepsis [42], arthritis [43] or gastritis [44]. In addition, the same dose of OLE is reported to reduce the LDL/HDL ratio [45] and exert protective effects in experimental models of cerebral ischemia [45,46], as well as to prevent type-2 diabetes in rats by decreasing the total antioxidant capacity in the plasma and the expression superoxide dismutase in the liver [47]. 

Our results show that OLE treatment to old Wistar rats for 3 weeks induces significant changes in body composition, increasing gastrocnemius weight and the amount of interscapular brown adipose tissue. However, visceral adiposity was not affected by OLE administration. This result disagrees with previous studies that have reported a positive effect of OLE decreasing visceral adiposity in rodents both in vivo [48,49,50,51] and in vitro [52]. These discrepancies may be due to differences in species (rats vs mice), extract compositions or dosages. Moreover, the different results may also be due to differences in the experimental models used since most of the studies reporting the antiadipogenic effect of OLE in vivo were performed in models of diet-induced obesity [48,50,51]. Although increased adiposity is a common feature in aging and obesity [53], it is reported that the physiopathological mechanisms implied in both conditions are different, with obesity-induced adiposity being associated with increased energy balance and with aging-induced adiposity being more related to impaired fat metabolism [54]. Thus, it is possible that the antiadipogenic effect of OLE is dependent on the mechanism of fat mass accrual. However, although we did not find a positive effect of OLE in adiposity or in the gene expression of enzymes involved in the processes of lipid uptake, lipogenesis, or lipolysis, such as LPL, FASn or HSL, OLE administration prevented the aging-induced increase in the circulating levels of LDL-cholesterol in the same cohort of rats [37] and prevented the aging-induced increase in the mRNA levels of PPAR-γ, a factor which plays a major role in adipogenesis. Likewise, other authors have reported a significant effect of OLE decreasing adipogenesis through the downregulation of PPAR-γ expression [50,51,52,55] and exerting anticholesterolemic effects in vitro [56] and in vivo, both in experimental animals [48,57,58,59,60] and in humans [61,62].

Since BAT amount and function are related to longevity [63,64], an important finding of this study is the positive effect of OLE increasing the amount of brown adipose tissue. This effect does not seem to be mediated by increased lipogenesis or increased lipid uptake since the mRNA levels of FASn and LPL are unchanged between untreated old rats and old rats treated with OLE. However, even though the gene expression of HSL was also unchanged between both experimental groups, OLE significantly downregulated the mRNA levels of β3 adrenoreceptor and CPT-1a, suggesting that the increased BAT mass may be the result of decreased lipolysis and fatty acid oxidation. However, since the mRNA levels of UCP-1 were significantly downregulated in OLE-treated rats, our results show that the increased BAT adiposity was not associated with increased UCP-1 mediated thermogenesis. Thus, the physiological relevance of increased BAT after OLE administration to aged rats remains to be determined and requires further investigations.

Our results also show that OLE attenuates the aging-induced decrease of gastrocnemius weight and protein content, exerting a positive effect on sarcopenia. This is a major finding since accentuated muscle loss in older adults is associated with a higher mortality risk compared to those with relative muscle preservation, suggesting that conservation of muscle mass is important for survival in old age [65]. This result agrees with previous studies that reported the positive effects of polyphenols from both EVOO and OLE preventing muscle atrophy in an experimental model of early osteoarthritis [66]. In addition, OLE supplementation to fish is reported to exert hypertrophic effects on muscle mass by increasing the myofibril and collagen content [67]. The protective effect of OLE on age-induced muscle loss seems to be mediated, at least in part, by its anti-inflammatory effects since both the circulating levels [37] and the mRNA levels of IL-6 in the gastrocnemius muscle were significantly reduced in aged rats treated with OLE compared to old untreated rats. Furthermore, OLE significantly increased the gene expression of the anti-inflammatory cytokine IL-10 that was reduced in old untreated animals. Thus, OLE treatment significantly reduces the IL-6/IL-10 ratio in aged rats, which has been used as a biomarker of sarcopenia in humans [68].

Another possible mechanism that may mediate the positive effects of OLE on aging-induced muscle loss is the decrease in HDAC-4, a protein that is overexpressed in sarcopenic muscles [15,16,17]. Moreover, OLE treatment prevented the aging-induce increase in the gene expression of myogenin, a protein regulated by HDAC-4 [69] that plays a key role in muscle atrophy through the activation of atrogenes [70]. Likewise, a recent study published by our group showed how a mixture between algae and EVOO oils also prevented sarcopenia in aged rats, an effect that was associated with a decrease in the gene expression of both HDAC-4 and myogenin in the gastrocnemius muscle [15]. Moreover, OLE administration prevents the aging-induced increase in the mRNA levels of myostatin, a protein that is overexpressed in sarcopenic muscles and decreases the muscle regenerative capacity [71]. 

Since polyphenols are present both in EVOO and in OLE, it is possible that these molecules are responsible, at least in part, for the beneficial effects of both nutraceuticals on aging-induced sarcopenia. Indeed, flavonoids have been proposed as potential candidates to treat muscle atrophy due to their role in improving mitochondrial function [72] and decreasing proteolysis through the downregulation of myostatin [73] and the atrogenes MURF-1 and atrogin-1 [74]. Moreover, flavonoids are reported to promote myogenesis, upregulating the protein expression of myogenic markers, such as MyoD and myogenin [74]. In this regard, our results show that aging was associated with an upregulation of both MyoD and myogenin and that OLE treatment significantly reduced their mRNA expression in the gastrocnemius muscle. Since it is reported that muscle wasting and muscle atrophy are usually associated with increased expression of markers involved in muscle regeneration as a compensatory mechanism [75,76], the positive effect of OLE preventing the aging-induced overexpression of both myogenic markers may be related to a better muscle condition in the treated animals (Figure 8).

Finally, another major finding of this study is that OLE treatment to aged rats for three weeks improves insulin sensitivity by increasing the gene expression of IR and/or GLUT-4 and activating the PI3K/Akt pathway both in visceral adipose tissue and skeletal muscle. These results may be related to the increased circulating adiponectin levels previously reported in the same cohort of rats [37] and are in agreement with a recent study that has reported the positive effects of polyphenols from OLE on muscle glucose homeostasis [78]. The insulin-sensitizing effects of OLE in visceral adipose tissue and skeletal muscle have already been reported in other tissues such as the liver and the aorta in the same cohort of rats [37] and seem to be mediated, at least in part, by its anti-inflammatory effect since OLE administration significantly downregulates the mRNA levels of IL-1β in epidydimal adipose tissue and the gene expression of COX-2, IL-6 and IL-1β in the gastrocnemius muscle. Moreover, treatment with OLE prevented the aging-induced downregulation of IL-10 mRNA levels in both tissues. These results agree with those reported by other authors in which OLE attenuates the gene expression of proinflammatory markers in visceral adipose tissue in experimental models of diet-induced obesity [51] and type-1 diabetes [77]. However, the gene expression of antioxidant enzymes was unaffected by OLE treatment in both tissues suggesting that the positive effects of OLE improving insulin sensitivity are more related to its anti-inflammatory rather than to its antioxidant effects. On the contrary, we have reported in a previous study that OLE prevents the aging-induced decrease in the gene expression of GSR, GPx and SOD-1 in the liver [37], suggesting that the antioxidant effects of OLE are tissue-specific. Likewise, OLE administration to aged rats was able to decrease oxidative stress in the liver but not in other organs such as the heart or the brain [41], suggesting that the liver is much prone to the antioxidant effects of OLE than other organs.

The effect of OLE increasing insulin sensitivity in skeletal muscle may be related, at least in part, to its positive effect on sarcopenia since both phenomena are intimately related. Indeed, it is reported that there is a positive correlation between insulin sensitivity and muscle mass in aged rats [29]. Moreover, blockade of sarcopenic markers, such as myostatin, not only prevents sarcopenia but also increases insulin sensitivity in old mice [30].

## 5. Conclusions

In conclusion, OLE administration to aged rats increases insulin sensitivity in adipose tissue and skeletal muscle and attenuates sarcopenia by decreasing the gene expression of proinflammatory cytokines and muscle atrophy markers. Thus, it may be a good candidate for the treatment and/or prevention of muscle loss in aged individuals.

## Figures and Tables

**Figure 1 antioxidants-10-00737-f001:**
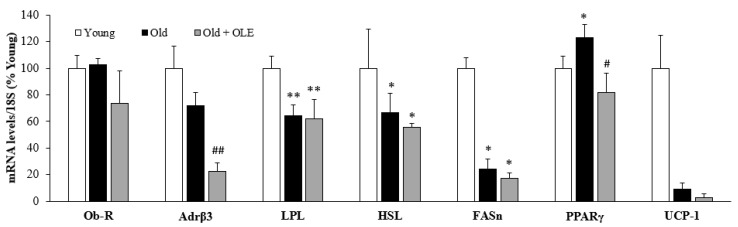
Gene expression of leptin receptor, β3 adrenergic receptor, lipoprotein lipase, hormone-sensitive lipase, fatty acid synthase, peroxisome proliferator-activated receptor-γ and un-coupling protein 1 in the epidydimal white adipose tissue of young rats, old rats and old rats treated 21 days with the OLE. Values are represented as mean ± SEM.* *p* < 0.05 vs. Young; ** *p* < 0.01 vs. Young; # *p* < 0.05 vs. Old; ## *p* < 0.01 vs. Old.

**Figure 2 antioxidants-10-00737-f002:**
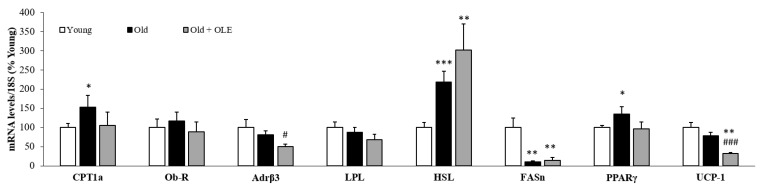
Gene expression of carnitine palmitoyltransferase 1a, leptin receptor, β3 adrenergic receptor, lipoprotein lipase, hormone-sensitive lipase, fatty acid synthase, peroxisome proliferator-activated receptor-γ and un-coupling protein 1 in the subscapular brown adipose tissue of young rats, old rats and old rats treated 21 days with the OLE. Values are represented as mean ± SEM.* *p* < 0.05 vs. Young; ** *p* < 0.01 vs. Young; *** *p* < 0.001 vs. Young; # *p* < 0.05 vs. Old; ### *p* < 0.001 vs. Old.

**Figure 3 antioxidants-10-00737-f003:**
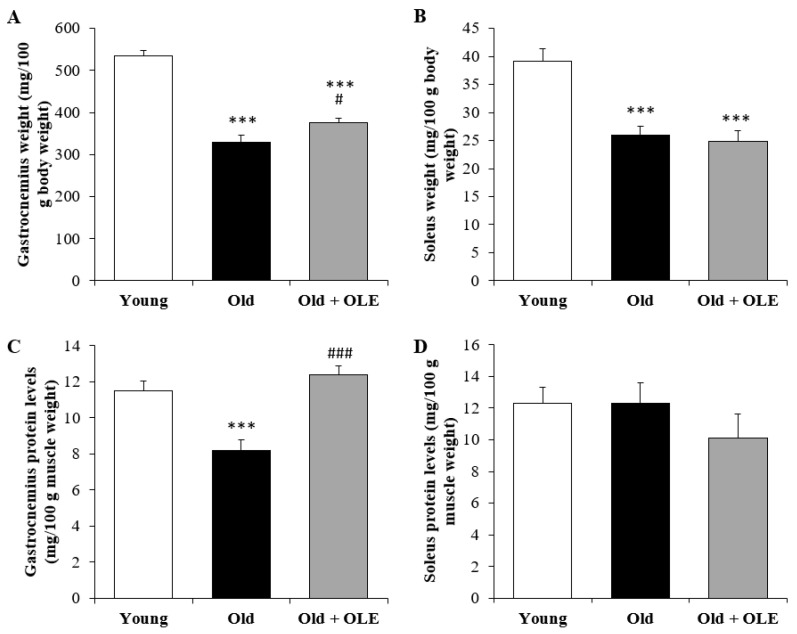
Relative muscle weight of gastrocnemius (**A**) and soleus (**B**), and protein levels of the gastrocnemius (**C**) and soleus (**D**) of young rats, old rats and old rats treated 21 days with the OLE. Values are represented as mean ± SEM. *** *p* < 0.001 vs. Young; # *p* < 0.05 vs. Old; ### *p* < 0.001 vs. Old.

**Figure 4 antioxidants-10-00737-f004:**
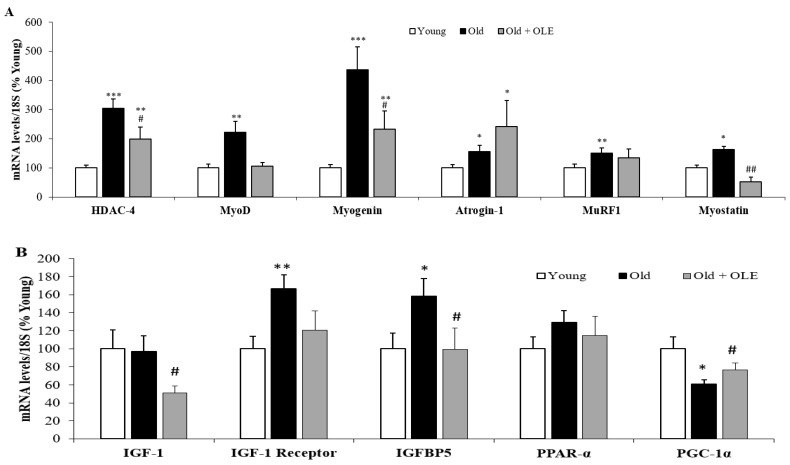
Gene expression of histone deacetylase 4, myoblast determination protein 1, myogenin, atrogin-1, muscle RING-finger protein-1 and myostatin (**A**), and insulin growth factor 1, IGF-1 receptor, IGF-binding protein 5, peroxisome proliferator-activated receptor α and PPAR-γ coactivator 1α (**B**) in the gastrocnemius of young rats, old rats and old rats treated 21 days with the OLE. Values are represented as mean ± SEM.* *p* < 0.05 vs. Young; ** *p* < 0.01 vs. Young; *** *p* < 0.001 vs. Young; # *p* < 0.05 vs. Old; ## *p* < 0.01 vs. Old.

**Figure 5 antioxidants-10-00737-f005:**
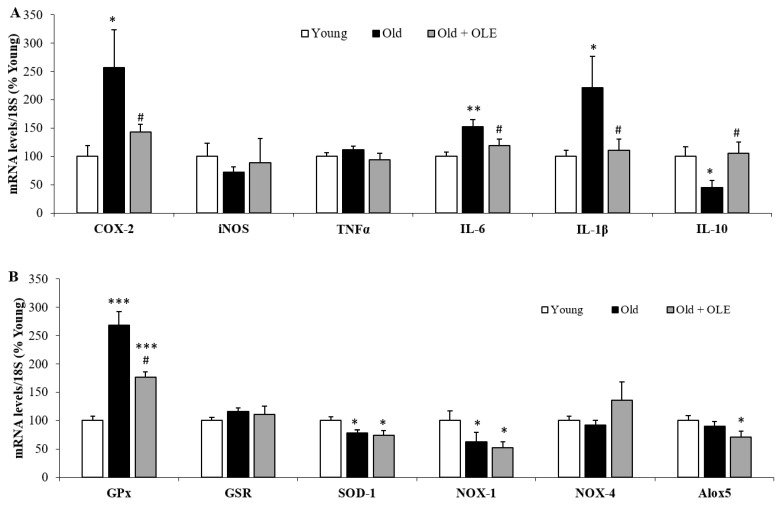
Gene expression of cyclooxygenase-2, inducible nitric oxide synthase, tumor necrosis factor α, interleukin 6, interleukin 1β and interleukin 10 (**A**), and glutathione peroxidase, glutathione reductase, super oxide dismutase 1, NADPH oxidase 1 and 4 and lipoxygenase (**B**) in the gastrocnemius of young rats, old rats and old rats treated 21 days with the OLE. Values are represented as mean ± SEM.* *p* < 0.05 vs. Young; ** *p* < 0.01 vs. Young; *** *p* < 0.001 vs. Young; # *p* < 0.05 vs. Old.

**Figure 6 antioxidants-10-00737-f006:**
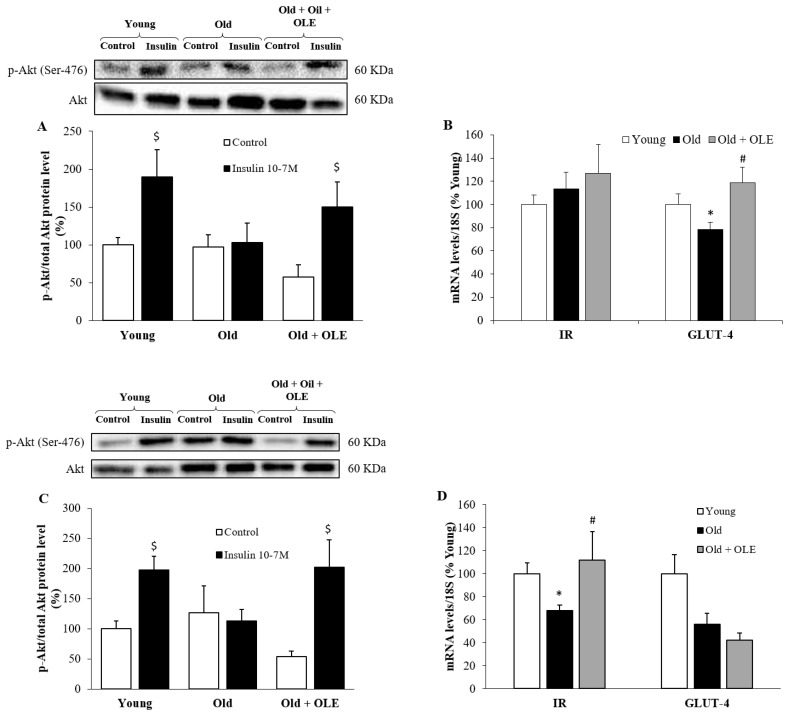
The protein ratio of p-Akt and Akt in the presence/absence of insulin 10^−7^ M (**A**) and mRNA concentrations of Insulin receptor and GLUT-4 (**B**) in gastrocnemius explants, and protein ratio of p-Akt and Akt in the presence/absence of insulin 10^-7^M (**C**) and mRNA concentrations of Insulin receptor and GLUT-4 (**D**) in epididymal visceral adipose tissue explants of young rats, old rats and old rats treated 21 days with the OLE in vitro. Values are represented as mean ± SEM. $ *p* < 0.05 vs. Control; * *p* < 0.05 vs. Young; # *p* < 0.05 vs. Old.

**Figure 7 antioxidants-10-00737-f007:**
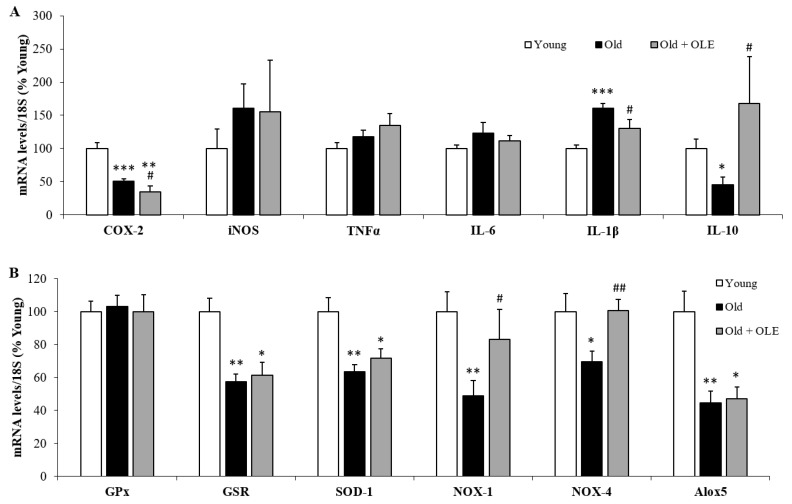
Gene expression of ciclooxigenase-2, inducible nitric oxide synthase, tumor necrosis factor α, interleukin 6, interleukin 1β and interleukin 10 (**A**), and glutathione peroxidase, glutathione reductase, super oxide dismutase 1, NADPH oxidase 1 and 4 and lipoxygenase (**B**) in the epidydimal white adipose tissue of young rats, old rats and old rats treated 21 days with the OLE. Values are represented as mean ± SEM.* *p* < 0.05 vs. Young; ** *p* < 0.01 vs. Young; *** *p* < 0.001 vs. Young; # *p* < 0.05 vs. Old.

**Figure 8 antioxidants-10-00737-f008:**
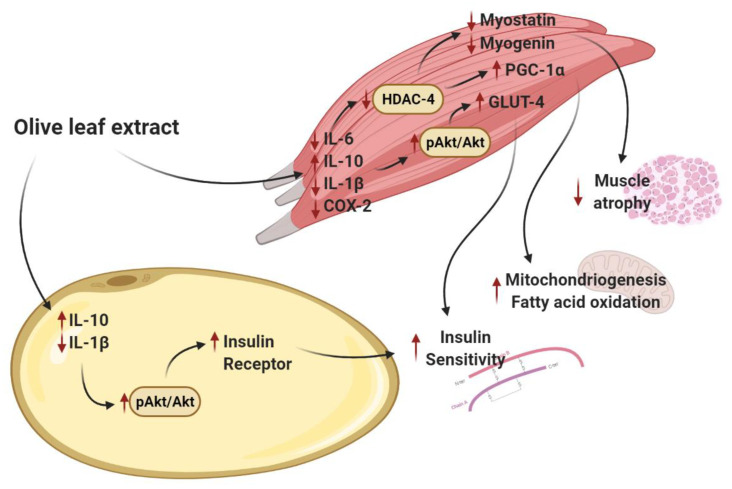
Schematic representation of the effects of the olive leaf extract on the main aging-induced alterations in white adipose tissue and skeletal muscle. OLE treatment decreases the inflammatory status in skeletal muscle, which results in a reduction of sarcopenia through a decrease in the gene expression of HDAC-4 and its effectors, myogenin [69,70] and myostatin [71]. Moreover, decreased inflammation is also associated with increased mitochondriogenesis and fatty acid oxidation in skeletal muscle [72] and with improved insulin sensitivity both in skeletal muscle and in visceral adipose tissue through a positive effect in the activation of the PI3K/Akt pathway [51,77]. GLUT-4 = glucose transporter 4; HDAC-4 = histone deacetylase 4; IL = interleukin; PGC-1α = peroxisome proliferator-activated receptor-gamma coactivator 1α.

**Table 1 antioxidants-10-00737-t001:** Relative organ weights of young rats, old rats and old rats treated with olive leaves extract (OLE).

	Young	Old	Old + OLE
Heart (mg/100 g body weight)	314.6 ± 10.8	291.0 ± 14.4	264.6 ± 13.7 *
Epidydimal visceral adipose tissue (mg/100 g body weight)	2107.1 ± 183.7	3201.8 ± 139.6 ***	2970.7 ± 265.1 **
Lumbar subcutaneous adipose tissue (mg/100 g body weight)	1063.1 ± 139.6	3974.2 ± 661.4 ***	4249.9 ± 396.9 ***
Interscapular brown adipose tissue (mg/100 g body weight)	105.6 ± 9.8	111.1 ± 12.9	140.4 ± 14.4 *#
Periaortic adipose tissue (mg/100 g body weight)	39.3 ± 5.9	47.8 ± 7.5	55.2 ± 5.9 *
Liver (mg/100 g body weight)	2932.8 ± 103.6	2204.2 ± 138.6 ***	2343.0 ± 92.6 **

Data are represented as mean value ± SEM; *n* = 6–11 samples/group. * *p* < 0.05 vs. Young; ** *p* < 0.01 vs. Young; *** *p* < 0.001 vs. Young; # *p* < 0.05 vs. Old.

## Data Availability

Data is contained within the article.

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
