# Peer review of "Olive Leaf Extract Supplementation to Old Wistar Rats Attenuates Aging-Induced Sarcopenia and Increases Insulin Sensitivity in Adipose Tissue and Skeletal Muscle"

_antioxidants, 2021, doi:10.3390/antiox10050737_

Round 1
Reviewer 1 Report
Comment and suggestion to authors:
Manuscript ID: antioxidants-1189800
Titled: "Olive leaf extract supplementation to old Wistar rats attenuates aging-induced sarcopenia and increases insulin sensitivity in adipose tissue and skeletal muscle"
- In this study, it is interesting that the authors conduct this work using leaves of olive. And the authors often emphasize that this is phenolic-rich olive extract, this would be clearly seen if the authors analyze and add the phenolic profiles of olive comparing between leaf and other part i.e., fruit, seed and stem.
- In the section “Introduction”, the authors should explain why the dose 100 mg/kg orally administration is suitable for this study, and add some supported references.
- In the Materials and Methods, section “ 2.1. Materials” the authors wrote that “extract composition has been previously described [37].” Indeed, a short summary of major composition of this extract should be provided, so as to help the readers to follow this manuscript easily.
- In the section “Discussion”, The more previous published works related this study should be added to discuss with these results.
- In Line 400-460, it would be very attractive and easy to understand for the readers, if the authors add some graphical diagrams to summarize their discussion.
- There are some spelling mistakes and grammatical error found in this manuscript, the author should pay attention on this point and check the whole manuscript before re-submission.
Reviewer 2 Report
González-Hedström et al. prepared a manuscript “Olive leaf extract supplementation to old Wistar rats attenuates aging-induced sarcopenia and increases insulin sensitivity in adipose tissue and skeletal muscle.” This study is well designed, methodology is appropriate to the objectives, and conclusions made are supported by the results. This was achieved by studying the mRNA concentrations of proteins/enzymes related to lipid metabolism, thermogenesis, sarcopenia, oxidative stress, inflammation, and glucose, as well as Western blotting of selected proteins. Overall conclusion is that llive leaf extract increases insulin sensitivity in adipose tissue and skeletal muscle and attenuates sarcopenia by decreasing the gene expression of proinflammatory cytokines and muscle atrophy markers.
Major drawback of the study is lack of determination of chemical constituents responsible for the observed biological effect.
Minor remark: O in “o-glucoside” is symbol for oxygen. Thus, it should be capitalized.
Reviewer 3 Report
The study of González-Hedström et al. examined the effect of the supplementation with an extract of olive leaves for 21 days on sarcopenia and insulin resistance in old Wistar rats. In general, this straightforward, highly useful study may help to address how old people can maintain their muscle mass and regulate their insulin sensitivity. This study is well-designed and all the experiments are thorough.
Round 2
Reviewer 1 Report
Comment and suggestion to authors:
Manuscript ID: antioxidants-1189800_Round II
Titled: "Olive leaf extract supplementation to old Wistar rats attenuates aging-induced sarcopenia and increases insulin sensitivity in adipose tissue and skeletal muscle"
- In Figure 8., it is nice, but some points in the diagram that the authors did not discovered by this work; the related references must be added.
- Some spelling mistakes and grammatical error still found in some sentences, the author should check the whole manuscript before re-submission.
